# Utilization of Carbide Slag by Wet Grinding as an Accelerator in Calcium Sulfoaluminate Cement

**DOI:** 10.3390/ma13204526

**Published:** 2020-10-13

**Authors:** Xianyue Gu, Hongbo Tan, Xingyang He, Olga Smirnova, Junjie Zhang, Zhongtao Luo

**Affiliations:** 1State Key Laboratory of Silicate Materials for Architectures, Wuhan University of Technology, Wuhan 430070, China; 290589@whut.edu.cn (X.G.); zhangjunjie@whut.edu.cn (J.Z.); 2School of Civil Engineering, Architecture and Environment, Hubei University of Technology, Wuhan 430070, China; hexycn@163.com; 3Department of Constructing Mining Enterprises and Underground Structures, Saint-Petersburg Mining University, 199106 Saint-Petersburg, Russia; smirnovaolgam@rambler.ru; 4School of Materials Science and Engineering, Zhengzhou University, Zhengzhou 450052, China; luozhongtaozz@163.com

**Keywords:** carbide slag, calcium sulfoaluminate cement, rapid setting, early strength

## Abstract

In this study, wet-ground carbide slag (i.e., WGCS) was utilized as an accelerator in calcium sulfoaluminate cement (CSA) for obtaining considerably faster setting processes for some special engineering processes such as plugging projects and rapid repair engineering. The WGCS–CSA system was designed, in which the replacement ratio of CSA by carbide slag was chosen as 4%, 8% and 12%. The setting time and compressive strength were measured, and the mechanism of the system hydration was studied in detail by means of calorimetry, XRD, thermogravimetry (TG) and SEM. The results showed that WGCS shortened the setting time of cement and significantly augmented the early strength. The addition of 8% of WGCS contributed to increasing the 2-h compressive strength from 4.2 MPa to 32.9 MPa. The decrease in the setting time and the increase in the initial strength were mainly attributed to the high initial pH value of the liquid phase and the high content of calcium ions in WGCS. Both these factors contributed to the ettringite formation and, at the same time, to the transformation of the morphology at a later time. Such results testify that WGCS can be used as an accelerator in the CSA system and also that it provides a novel approach to the reutilization of carbide slag.

## 1. Introduction

Calcium sulfoaluminate cement (CSA) is a commercially produced cement which is attracting more and more attention in rapid construction and repair projects due to its fast setting and high early strength [1,2,3,4]. Compared with Portland cement (PC), CSA has larger potential reductions in energy and greenhouse gas emissions because of its production at lower temperatures and the lower amount of calcium contained in its feedstock. As a rule, the setting time of CSA is less than 25 min according to Chinese National Standard GB 20472-2006 (sulfoaluminate cement). Despite the fact that the setting time is much shorter than that of PC, it turns out that CSA does not always meet the setting time requirement for repairing a special engineering system such as a plugging project. In Chinese National Standard GB 23440-2009 (Inorganic waterproof and leakage-preventing materials), the setting time was limited to 5 min. As reported in the literature, lithium salts such as lithium carbonate and lithium nitrate were able to efficiently promote the hydration process of CSA [5,6], and obtain considerably faster setting to meet that requirement. However, this kind of accelerator seemed to be too expensive for construction materials. The search for new kinds of accelerators for CSA is a relevant issue. This search can be performed among secondary resources, such as industrial byproducts [7,8,9]. The use of secondary resources for obtaining materials with improved properties corresponds to the principles of the circular economy as well as to the development of technologies that reduce carbon dioxide emissions [10,11,12].
C_4_A_3_S + 3CSH_2_ + 34H→AFt + 3AH_3_(1)
C_2_S + 3H→C–S–H + CH(2)
AH_3_(gel) + 3CH + 3CSH_2_ + 20H→AFt(3)
C_4_A_3_S + 18H→AFm + 3AH_3_(4)
C_4_A_3_S + 8CSH_2_ + 6CH + 74H→3AFt(5)

The fast setting of the CSA paste was closely related to its early hydration. In the literature, the hydration can be expressed as Equations (1)–(5) [13,14,15,16,17]. After being mixed with water, the calcium sulfoaluminate (Ye’elimite, C_4_A_3_S, 4CaO·3Al_2_O_3_·SO_3_) reacted with gypsum and water, and then ettringite (AFt, 3CaO·Al_2_O_3_·3CaSO_4_·32H_2_O) and AH_3_ was produced; AH_3_ gel further reacted with gypsum to produce AFt, as shown in Equation (3). If the gypsum is depleted, Equation (4) should be used. The fast formation of AFt mainly determined the setting process of CSA. From Equations (1), (3), and (5), it is clear that gypsum in CSA has an important role in the setting process and it is considered as one of the common accelerators for the early hydration of CSA [18,19]. The proper amount of gypsum in CSA could facilitate the formation of AFt and also notably shorten the setting time [20,21,22]. However, if an excessive dosage of gypsum is used it could produce some negative effects such as a potential risk of crack formation in the cement matrix [23]. Furthermore, the presence of lithium ions efficiently facilitated the early hydration of CSA [24]. Lithium ions in lithium slag caused the disappearance of the induction period in CSA hydration, leading to the setting time being shortened and the early strength of the hardened paste being significantly increased [25]. Lithium salts such as Li_2_SO_4_, LiOH and LiNO_3_ could obviously accelerate the CSA setting process [26,27]. The main reason was that these lithium ions could be precipitated with aluminum hydroxide (AH_3_) such as Li-Al(OH)_3_, which could accelerate the ettringite formation [28]. Moreover, nano compounds such as nano-SiO_2_ were good for promoting the early hydration of CSA [29]. It was reported that the use of nano-SiO_2_ shortened the setting time of belite–CSA pastes and CSA–PC pastes [30]. The early strength was also enhanced by the addition of nano-SiO_2_ [31,32,33]. The reason was that nanoparticles could act as crystal nuclei to induce the initial ettringite formation and their filling effect also refined the microstructure. The discussion described above helps us to understand the setting process of CSA.

Moreover, in the CSA system, the morphology of the formed AFt was easily influenced by the pH value of the pore solution. The length/diameter ratio of AFt crystals could be minimized by the elevated pH value [34] and the transformation of a needle-shaped AFt into a stick-shaped one would take place. Obviously, it was more likely that the needle-shaped ettringites would be interlocked together [35], leading to fast setting and a compact microstructure. It was reported that 3% calcium oxide improved the initial pH value of CSA from 10.8 to 11.6 and increased the content of AFt from 14.69% to 27.14% at the age of 2 h [36]. It was also demonstrated that the needle-shaped AFt could be easily formed in a CSA–PC–calcium aluminate cement ternary binder because of the presence of free lime in PC, offering a relatively high pH value [37]. It was well known that carbide slag (CS) was the solid waste emitted in the process of manufacturing the acetylene gas and its main ingredients were Ca(OH)_2_ and CaCO_3_ [38,39]. It was inferred that CS would accelerate the setting process of the CSA system because of the Ca(OH)_2_ contained in CS. Furthermore, the commercial cost of Li_2_CO_3_ and nano-SiO_2_ was approximately 43,600 and 23,000 China yuan (CNY)/t respectively, but the cost of CS was 50–120 CNY/t. It was noted that the cost of CS was much less in actual engineering applications than most commercial accelerators. The use of CS as an accelerator in the CSA system would most likely lead to great economic and environment benefits.

In the present study, an attempt to utilize CS to accelerate the setting process of CSA was undertaken, and raw CS (RCS) was treated by wet grinding to obtain finer particles (i.e., wet-ground carbide slag (WGCS)) [40]. The setting time and compressive strength of the CS–CSA binder were measured and the hydration process was studied by calorimetry, XRD, thermogravimetry (TG)–differential thermal gravity (DTG), and SEM. Finally, the mechanism was discussed in detail. The results suggested that CS could be used as an accelerator for CSA and, at the same time, provided a new approach to the utilization of CS.

## 2. Materials and Methods

### 2.1. Materials

CSA (425 type), in accordance with the Chinese National Standard GB 20472-2006, was used in this study. Carbide slag (CS) emitted from the acetylene gas production plant was also used. The chemical compositions of CSA and CS were evaluated by X ray Fluorescence (XRF) and the results are listed in Table 1.

### 2.2. Preparation of Specimens

#### 2.2.1. Preparation of Wet-Ground CS (WGCS)

In order to promote the efficiency of wet grinding, the optimal parameters of wet grinding were obtained after three experiments, which are listed in Table 2. It was noted that the water to raw CS (RCS) ratio was 1:1. Commercially available polycarboxylate superplasticizer (PCE), with a solid content of 40%, was added in a quantity of 1.5% of RCS mass. Zirconia balls with four various diameters were used as the grinding media and their total mass was 300 g (their weight ratio was 10 mm:8 mm:5 mm:3 mm = 1:4:4:1) [41]. The process of wet grinding was as follows: the grinding media, water, RCS and dispersant were all mixed together in a self-made vertical stirred mill (VSM); after being milled for 60 min at a speed of 400 rpm, the zirconia balls were separated and WGCS slurry was obtained. The water loss in the grinding process was not considered but the water in the WGCS slurry was considered while preparing the samples.

The particle size distribution of CSA, RCS, and WGCS was characterized with a laser particle size analyzer (LPSA, Mastersizer 2000; Malvin Inc., London, UK), and the results are shown in Figure 1. It was found that the median particle size (D50) of CSA, RCS, and WGCS was 15.5 μm, 11.8 μm, and 2.9 μm, respectively (the maximum error was 2 μm), and the fineness of CS grinding was significantly increased after the wet grinding process. Moreover, in order to clearly compare the particle size of RCS and WGCS, the samples were glued to the base with conductive tape after drying in a vacuum at 40 °C and placed in a field emission scanning electron microscope to obtain the SEM images, which are shown in Figure 2. Much finer particles were seen in WGCS than in RCS. XRD patterns of RCS and WGCS are shown in Figure 3, where the main phases in CS were portlandite (Ca(OH)_2_, International Centre for Diffraction Data (ICDD) # 04-0733) and calcite (CaCO_3_, ICDD # 05-0586) [42]. Moreover, after the wet grinding treatment, the peak density of portlandite was decreased and one possible reason for this was that, during the grinding process, the crystal structure of portlandite was altered by the mechanical grinding forces [43]. It was summarized that, in the process of wet grinding, much finer particles of WGCS were obtained, which could be beneficial for the dissolution process and could participate in the hydration process of CSA.

#### 2.2.2. Samples Preparation

The binder compositions are shown in Table 3. The amounts of CS were 4%, 8%, and 12% of CSA, and the water-to-binder ratio (w/b) was 0.27. The water in the WGCS slurry was also considered. The preparation of CS–CSA pastes was made in the following ways: CSA, RCS or WGCS, water, and PCE (0.7% of pastes) were mixed and then stirred for 3 min at high speed. After that, the pastes were cast into 40 × 40 × 40 mm^3^ molds and cured under a temperature of 20 ± 1 °C and more than 95% R.H. At the age of 1 h, they were demolded and further cured under the same conditions until the testing age. Furthermore, the pastes used for micro-analysis were treated in the following ways: at the testing age, the hardened pastes were crushed into small pieces by a pestle and the hydration was stopped by immersing the pieces into absolute ethanol for 3 h and, after that, drying them in a vacuum at 40 °C for 24 h [44,45]. Small pieces were used for the SEM measurements. These small pieces were additionally ground in order to pass them through a 45 μm sieve for testing by XRD and TG.

### 2.3. Methods

#### 2.3.1. Setting Time and Compressive Strength

The measurement of the setting time of CS–CSA pastes was conducted according to Chinese National Standard GB/T 1346-2011 and the setting time was measured by a Vicat apparatus [46]. The test method is as follows: the screw was loosened every 30 s to make the test needle sink vertically into the paste. The initial setting state was reached at the time of the test needle sinking to 4 mm ± 1 mm from the bottom plate. The time from adding all raw materials to the water to the initial setting state is the initial setting time; after that, the test mold was immediately translated, turned over and measured every 1 min. The final setting state was reached at the time of the test needle sinking into 0.5 mm of paste. The time from adding all of the raw materials into the water to the final setting state is the final setting time. The test of the setting time was done three times for one sample.

Compressive strength of the hardened paste was tested by a compressive machine (TYE–300F; Changzhou Ruipin Precision Instrument Co., Ltd., Changzhou, China) with a load rate of 2.4 kN/s. For each group, three specimens were tested at the ages of 2 h, 3 d and 7 d. Statistical methods were used for processing the results.

#### 2.3.2. Hydration Heat

The hydration heat of the WGCS–CSA binder (8% of CS, 92% of CSA) was measured with an isothermal calorimeter (TAM AIR, C80; SETARAM company, Shanghai, China). Approximately 3.0 g of binder was used in this experiment. The data were automatically recorded by the apparatus during the first 24 h of hydration.

#### 2.3.3. X-ray Diffractometer (XRD)

XRD patterns of CS–CSA specimens were obtained by an X-ray diffractometer (Model D8 Advance; Berlin, Germany) with Cu (Kα) radiation. The 2θ range was from 5° to 60° with a scanning speed of 4°/min [47].

#### 2.3.4. Thermogravimetric (TG)

TG patterns of CS–CSA specimens were obtained by a comprehensive thermal analyzer (STA449F3; Netzsch Instrument Manufacturing Co., Ltd., Berlin, Germany). Approximately 15 mg of the specimens were placed into Al_2_O_3_ pots. The temperature rise ranged from room temperature up to 1000 °C, with a heating rate of 10 °C/min under an N_2_ atmosphere.

#### 2.3.5. Scanning Electron Microscope (SEM)

SEM images of CS–CSA specimens were obtained by a field emission scanning electron microscope (Zeiss Ultra Plus, Berlin, Germany) and a Schottky field emission electron gun was used [48,49]. The vacuum degree of the sample chamber was low. The magnification was 10–100,000 and the maximum number of pixels was 6144 × 4096.

## 3. Results and Discussion

### 3.1. Compressive Strength

The effect of CS on the compressive strength of CSA pastes was investigated and the results are shown in Figure 4. As shown in Figure 4a, at the age of 2 h, the compressive strength of the Blank (pure CSA) was 4.2 MPa and 4% of WGCS increased it to 10 MPa, i.e., by 138%. A further increase in amount of WGCS up to 8% and 12% showed that the strength was increased to 32.9 MPa and 39.7 MPa, i.e., by 683% and 845% compared with the Blank, which exceeded the standard compressive strength of CSA at 1 d, 30.0 MPa (National Standard GB 20472-2006). However, it was noted that WGCS reduced the compressive strength at the ages of 3 and 7 days. One reason was the fact that the actual content of CSA was decreased by the growth of WGCS. Moreover, the initial hydration products had a strong covering effect on the cement clinker, inhibiting the further hydration of CSA.

The effect of RCS and WGCS on compressive strength was studied, and the results are shown in Figure 4b. It was observed that both RCS and WGCS improved the compressive strength of CSA pastes at the age of 2 h, increasing it by 469% for RCS and 683% for WGCS. Obviously, WGCS exhibited a higher strength increase than RCS. The probable reason for that was the much finer particles in WGCS than in RCS, which took part in the hydration more intensively. Moreover, both RCS and WGCS reduced the compressive strength at the age of 3 d and 7 d in comparison with the Blank, but WGCS–CSA still showed a higher compressive strength than RCS–CSA.

It was concluded that CS exhibited efficient strength-enhancing effects in CSA at a very early age. With the curing age increasing, the compressive strength development slows down, but at an age of more than 3 days, the compressive strength becomes lower than that of the Blank.

### 3.2. Setting Time

The effect of CS on the setting time of CSA pastes was investigated and the results are shown in Table 4. It is seen that, in comparison with the Blank, 4.0% of WGCS significantly shortened the initial setting time of CSA paste from 59 min to 12 min and the final setting time was also shortened from 71 min to 14 min. This result indicates that WGCS addition accelerates the setting process of CSA pastes. However, the increase in the WGCS amount slightly prolongs rather than shortens the initial and final setting times. Compared with the sample with 4% of WGCS, 12% of WGCS prolonged the initial setting time from 12 min to 16 min and the final setting time from 14 min to 19 min. Furthermore, it was noted that the effect of RCS and WGCS on the setting time had no obvious difference, and it might be related to the test measurement period (30 s). Moreover, at a low w/b, the effect of powder fineness on the setting time might be weakened by the reaction between CSA and CS.

It was confirmed that CS accelerated the setting process of the CSA system. The fast setting of the CS–CSA binder is well suited to rapid repair engineering and plugging projects.

### 3.3. Hydration Heat

Figure 5 shows the hydration heat of CS–CSA pastes. As shown in Figure 5, two exothermic peaks were observed, which were closely related to the formation of AFt [50,51]. The first one was caused by the reaction between Ye’elimite and gypsum, leading to AFt formation, as shown in Equation (1) [52]. At this stage, one AFt layer was formed on the surface of cement particles that could hinder further hydration. Over time, the AFt layer began to be ruined and then the second peak occurred. These two steps were repeated and the hydration of CSA continued [53]. In Figure 5a, it was found that both RCS and WGCS had no obvious effect on the first peak. However, the intensity of the second peak was increased. The reason was the fact that the formation of AFt was accelerated, as shown in Equation (5), leading to the rapid setting and high early strength of the hardened WGCS–CSA pastes. In Figure 5b, it was clearly seen that the hydration heat of WGCS–CSA was higher than that of the Blank within the first 6 h, but lower than that of the Blank afterwards, which was due to the strong covering effect caused by many AFt layers on the surface of CSA. Moreover, it was noted that the hydration heat of WGCS–CSA was always higher than that of RCS–CSA. According to Table 5, the addition of WGCS significantly promoted the release of hydration heat over 2 h. This illustrated that the hydration of CSA within 2 h was greatly accelerated by adding WGCS, which was one of the reasons for the fast setting of WGCS–CSA.

Thus, it may be concluded that WGCS greatly contributed to the release of the hydration heat of CSA at a very early period and this was the main reason for the much higher early compressive strength and much faster setting process of WGCS–CSA.

### 3.4. Hydration Analysis

#### 3.4.1. XRD Analysis

XRD patterns of WGCS–CSA paste samples are shown in Figure 6. In Figure 6a, Ye’elimite (C_4_A_3_S, ICDD # 33-0256), belite (C_2_S, ICDD # 33-0302) and anhydrite (CS, ICDD # 37-1496) can be clearly observed at the age of 2 h. The main hydrate, namely ettringite (AFt, ICDD # 41-1451), was also clearly found. It was noted that the peak intensity of AFt was noticeably increased by the addition of WGCS. The decrease in the peak intensity for Ye’elimite and anhydrite also indirectly indicated the AFt formation. It was also responsible for the higher early compressive strength. As shown in Figure 6b, the peak intensity of AFt for the Blank was much higher than that of WGCS–CSA pastes at the age of 3 days. The main reason was due to the lower content of CSA in WGCS–CSA. Additionally, from a 2-h age to a 3-day age, in both the Blank sample and WGCS–CSA sample, the peak intensity of AFt was increased, but Ye’elimite and anhydrite peak intensity was reduced because of the continuing hydration of CSA. This could also explain the compressive strength development in the period from 2 h to 3 days.

#### 3.4.2. TG–DTG Analysis

TG–DTG patterns of WGCS–CSA pastes are shown in Figure 7. It was noted that the mass loss of specimens was mainly focused on temperatures of about 100 °C, 250 °C, and 650 °C. This corresponds to the decomposition of AFt, monosulfide hydrated calcium sulfoaluminate (AFm, 3CaO·Al_2_O_3_·CaSO_4_·12H_2_O), and calcite, respectively [54]. As shown in Figure 7a, the peak intensity of the mass loss at a temperature of about 100 °C for the Blank at the age of 2 h was much lower than that for WGCS–CSA pastes, indicating that WGCS accelerated the AFt formation in the early period. The mass loss peak intensity at a temperature of about 650 °C was obvious in WGCS–CSA pastes due to the presence of CaCO_3_ in WGCS. In the period from 2 h of age to 3 days of age, the mass loss peak intensity representing AFt and AFm was increased due to the continuous hydration of CSA. In Figure 7b, the mass loss peak intensity representing AFt in the Blank sample exhibited no obvious difference from that of the WGCS–CSA sample at the age of 3 days.

The content of AFt in specimens was calculated by Equation (6) and the results are listed in Table 6 [36].
(6)$$\mathrm{AFt}\;\left(\%\right)=(\frac{M_{50}{-M}_{150}}{M_{\mathrm{total}}})/0.35$$
where M_50_ and M_150_ are the mass loss of specimens at temperatures of 50 °C and 150 °C; M_total_ is the total initial of specimens.

From Table 6, it is seen that the AFt content for the Blank was 14.9% at 2 h of age, while that for WGCS–CSA pastes was 26.2%. Thus, the increase was 75.8%. This indicates that WGCS accelerated the AFt formation in CSA in the early period. It was found that the content of AFt in specimens at the age of 3 days was decreased by adding the WGCS. On the one hand, the fast formation of the initial phase of AFt in the WGCS–CSA system led to the formation of crusts on the surface of cement clinker particles, thereby retarding the further hydration of CSA. On the other hand, WGCS replaced some part of CSA and the total amount of hydration products was reduced.

#### 3.4.3. SEM Analysis

SEM images of CS–CSA hardened pastes at the ages of 2 h and 3 days are shown in Figure 8. As shown in Figure 8a,b, stick-shaped AFt was observed in CSA paste at the age of 2 h, whereas needle-shaped AFt was found in the WGCS–CSA sample. It was reported in the literature that the addition of OH ions would increase the pH value in CSA, but the slight increase in pH value would reduce the aspect ratio of Aft, leading to a more compact structure in the CSA matrix [34].

Moreover, it was observed that AFt in CSA hardened pastes had a large aspect ratio, but the microstructure of the matrix was loose, as shown in Figure 8a. However, as shown in Figure 8b, the microstructure of the WGCS–CSA matrix was dense since the needle-shaped AFt crystals could easily interlace with one another. This was one of the reasons for the high early strength of the WGCS–CSA matrix. Furthermore, as shown in Figure 8c,d, short rod-shaped AFt was found in CSA hardened pastes, whereas a large amount of needle-shaped AFt was observed in WGCS–CSA hardened pastes. The difference between these two shapes was obvious. This illustrates the fact that, during a 3-day aging process, the microstructure of AFt can also be altered by WGCS.

Based on the discussion above, it was confirmed that WGCS accelerated the hydration of CSA pastes in the early period and also contributed to the formation of needle-shaped AFt.

### 3.5. Discussion

From what has been said above, it might be assumed that CS accelerated the setting of CSA and significantly augmented the early strength of CSA. It can be seen that the compressive strength of samples aged for 2 h increased by more than six times with 8% of WGCS. It is well known that a high early strength is essential for the rapid repair of concrete structures. Despite the much lower CSA content in the WGCS–CSA system, its compressive strength at the age of 2 h was several times higher than that of the CSA system. This also illustrates the fact that WGCS significantly promoted the early hydration of the CSA system. There are various reasons for this.

The hydration mechanism of CSA in the presence and absence of WGCS is shown in Figure 9. In the hardening of the CSA system at the age of 2 h, a stick-shaped AFt was formed, creating the loose matrix structure. However, in the WGCS–CSA system, owing to the high content of Ca(OH)_2_ in WGCS, the initial pH value in CSA paste was considerably higher and this contributed to the ettringite formation [37]. Moreover, the initial concentration of calcium ions in CSA paste was also increased and this also contributed to the ettringite formation. Due to the rapid ettringite formation, a dense crystal skeleton was formed, which led to fast setting and high early strength [36]. Moreover, it was noted that, in the presence of WGCS, needle-shaped AFt was formed and it seemed likely that this form of AFt compacted the microstructure, which, in turn, resulted in the early strength development.

### 3.6. Cost Assessment

It was confirmed that WGCS could be used as an accelerator for CSA early hydration as the early compressive strength was increased in the WGCS–CSA system. Comparative cost assessments of WGCS–CSA and CSA + Li_2_CO_3_ systems were performed.

In Table 7, the costs of WGCS, Li_2_CO_3_, and CSA are listed, and the costs of the WGCS–CSA system and CSA + Li_2_CO_3_ system are shown in Table 8. As reported in the literature, lithium carbonate (Li_2_CO_3_) was used as an accelerator to promote the setting process of CSA [55,56,57]. The cost of CSA was around 850 CNY/t (Tangshan Polar Bear Building Materials Co., Ltd., Tangshan, China) and the cost of Li_2_CO_3_ was around 43,600 CNY/t (Qinghai Lithium Co. Ltd., Golmud, China). If 0.03% of Li_2_CO_3_ was added into CSA as an accelerator, the initial setting time of CSA would be reduced to 30.0% and the final setting time to 28.0% [58]. The cost of the system was about 863 CNY/t. However, in the WGCS–CSA system, if 8% of WGCS was used to replace CSA, the initial and final setting times would be reduced to 23.7% and 22.5%, respectively, and the cost of the WGCS–CSA system would be about 790 CNY/t, which is lower than the cost of the CSA + Li_2_CO_3_ system. It seemed that WGCS showed great potential to be used as a chemical accelerator in CSA systems.

## 4. Conclusions


CS greatly shortened the setting time and increased the 2 h compressive strength of CSA, and we noted that 8% WGCS increased the 2 h compressive strength by more than six times.The higher early strength and fast setting of the CSA–CS system were explained by the fact that WGCS greatly facilitated ettringite formation and induced the transformation of stick-shaped ettringite to needle-shaped ettringite at an early age. This kind of ettringite seemed advantageous for shortening the setting time and accelerating the strength development.In contrast to the commonly used chemical accelerators, such as lithium carbonate, WGCS has a much lower cost. Such a result not only suggests another new method for the utilization of CS, but also offers a waste-based accelerator for the CSA system. It should be underlined that this study emphasized the early strength of the CSA–WGCS system, but its later strength development was not discussed.


## Figures and Tables

**Figure 1 materials-13-04526-f001:**
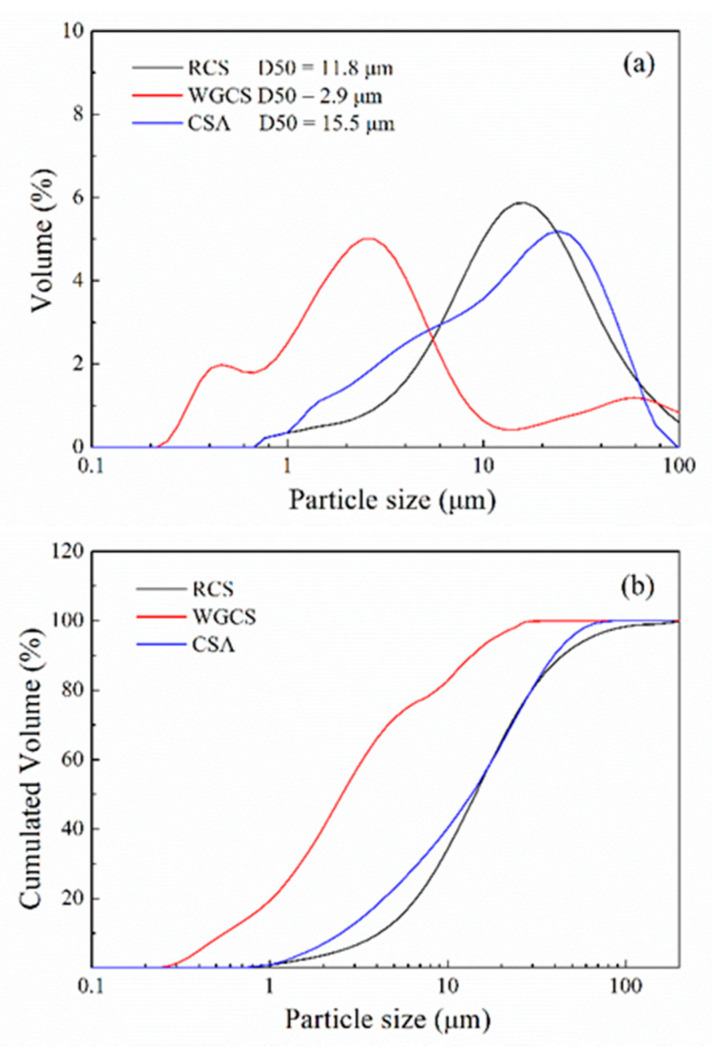
Particle size distribution of CSA, raw CS (RCS), and WGCS. (**a**) Volume; (**b**) Cumulated volume.

**Figure 2 materials-13-04526-f002:**
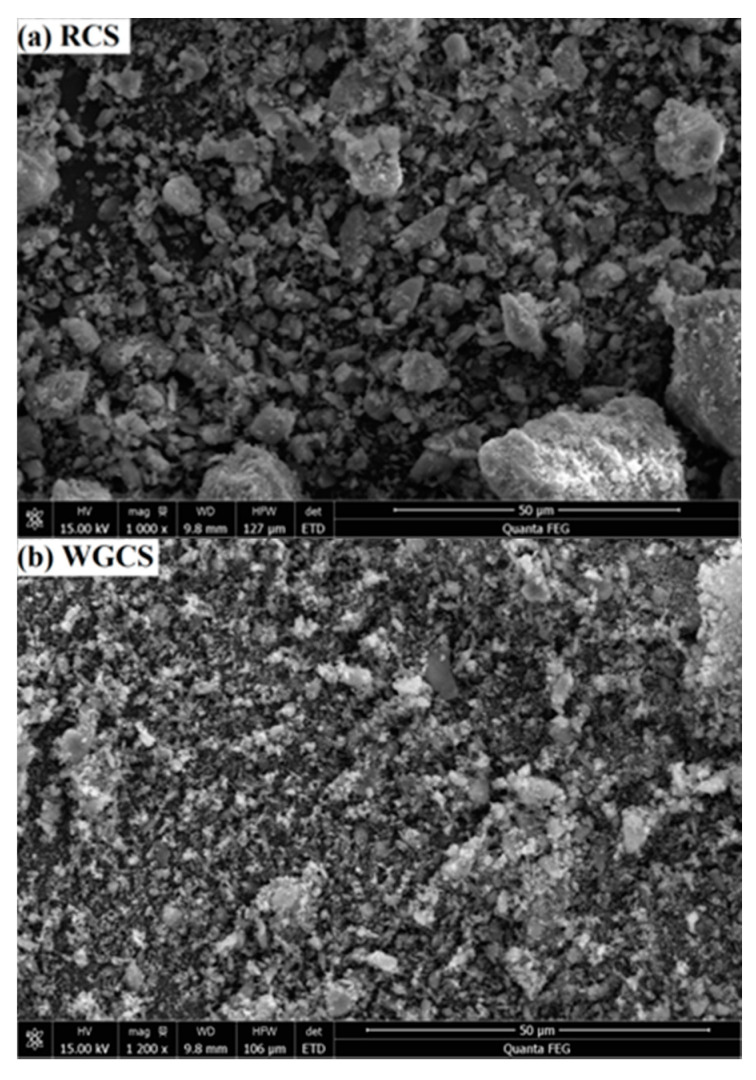
SEM images of RCS and WGCS. (**a**) RCS; (**b**) WGCS.

**Figure 3 materials-13-04526-f003:**
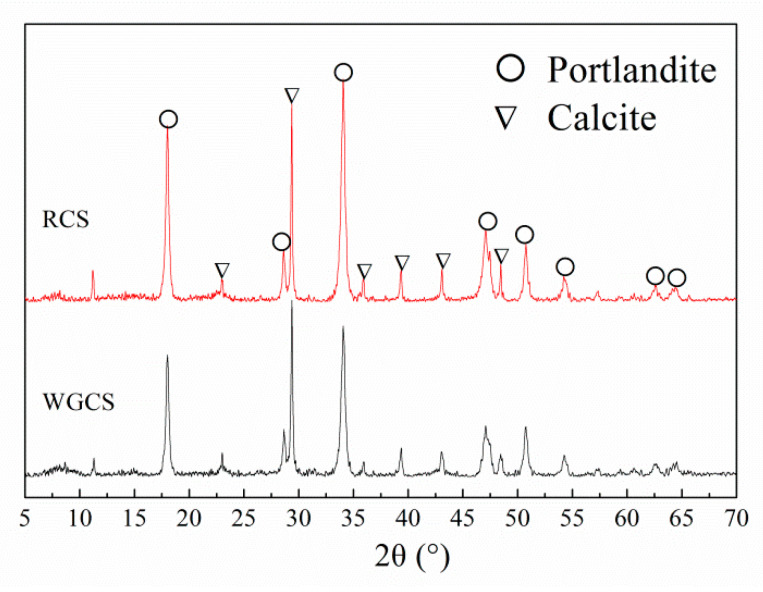
XRD patterns of RCS and WGCS.

**Figure 4 materials-13-04526-f004:**
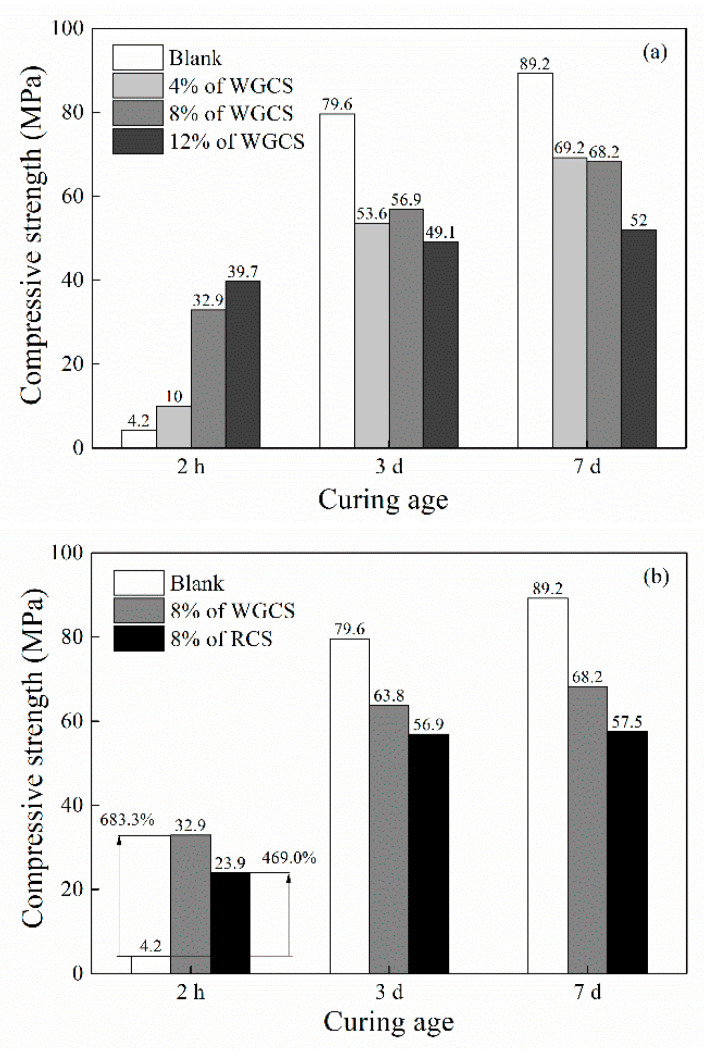
Compressive strength of CS–CSA pastes (maximum error was 5.4 MPa). (**a**) Blank and 4, 8, 12% of WGCS at 2 h, 3 d, 7 d; (**b**) Blank, 8% of RCS and 8% of WGCS at 2 h, 3 d, 7 d.

**Figure 5 materials-13-04526-f005:**
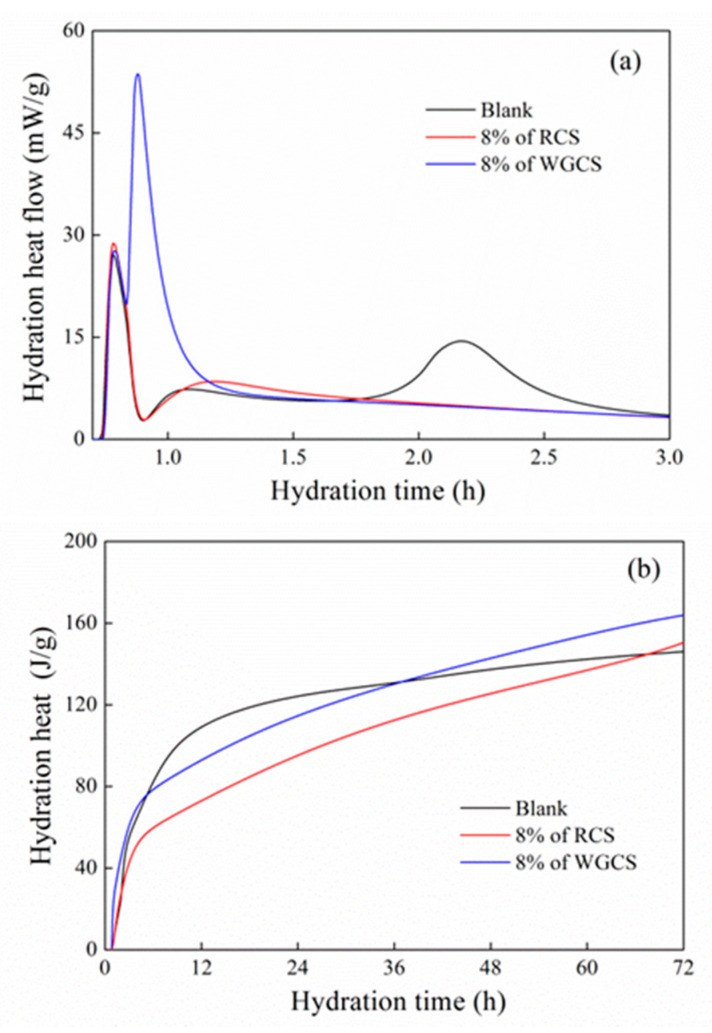
Hydration rate and heat of CS–CSA pastes. (**a**) Hydration heat flow; (**b**) Hydration heat.

**Figure 6 materials-13-04526-f006:**
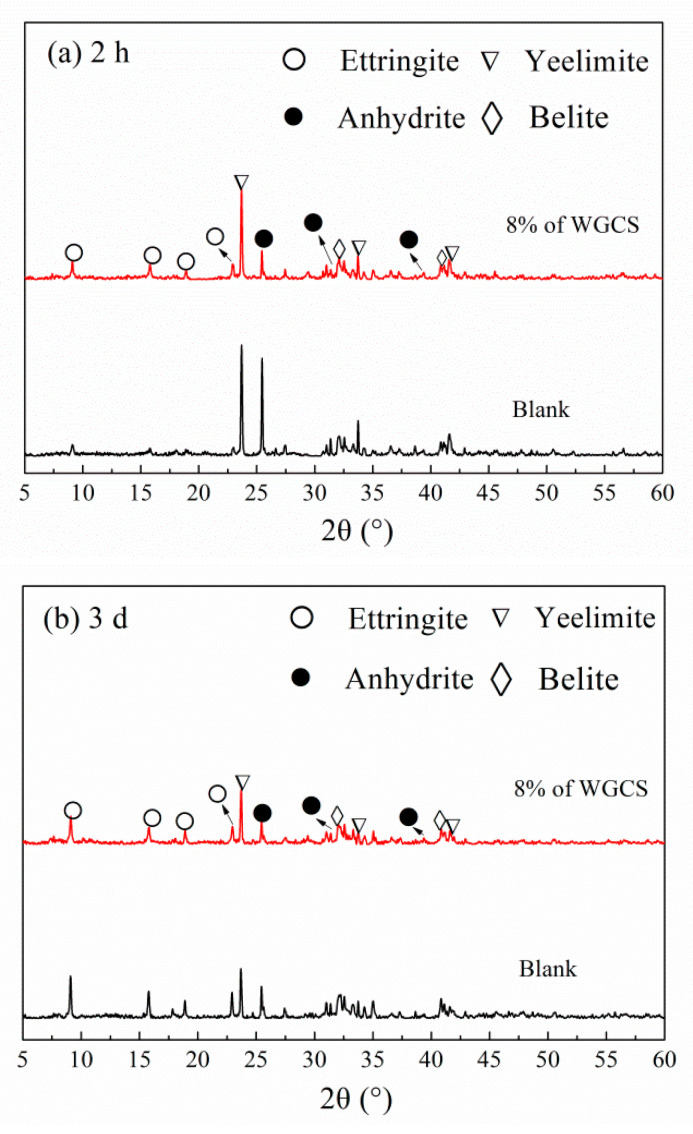
XRD patterns of WGCS–CSA paste. (**a**) 2 h; (**b**) 3 d.

**Figure 7 materials-13-04526-f007:**
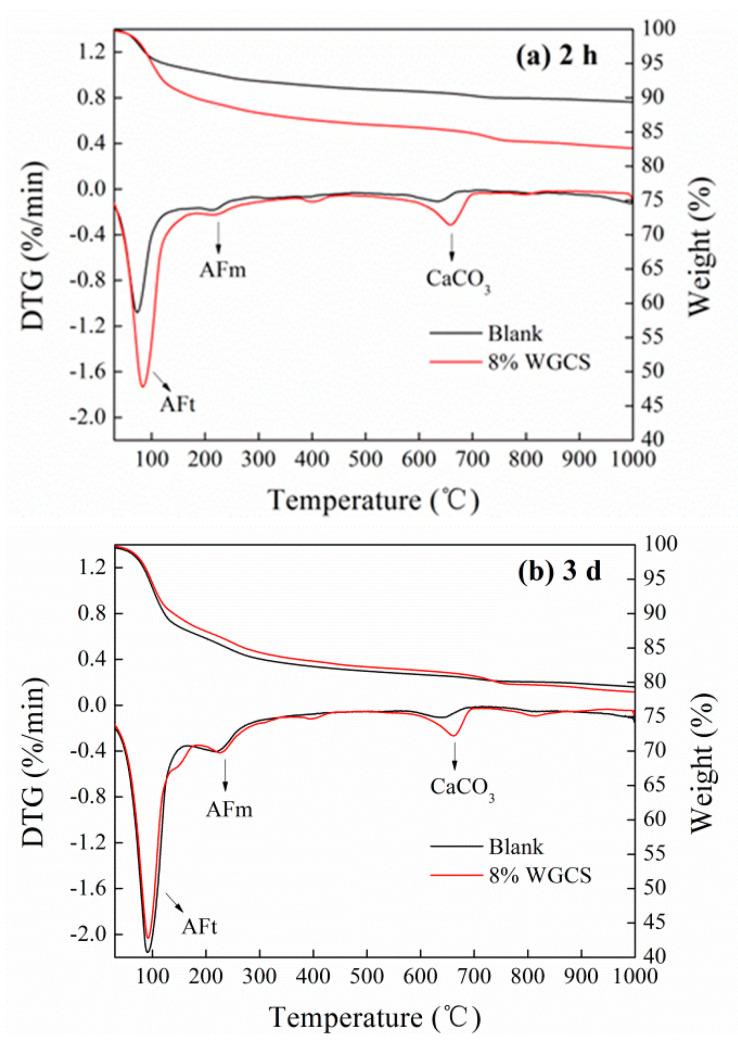
Thermogravimetry (TG)–Differential thermal gravity (DTG) patterns of CS–CSA paste. (**a**) 2 h; (**b**) 3 d.

**Figure 8 materials-13-04526-f008:**
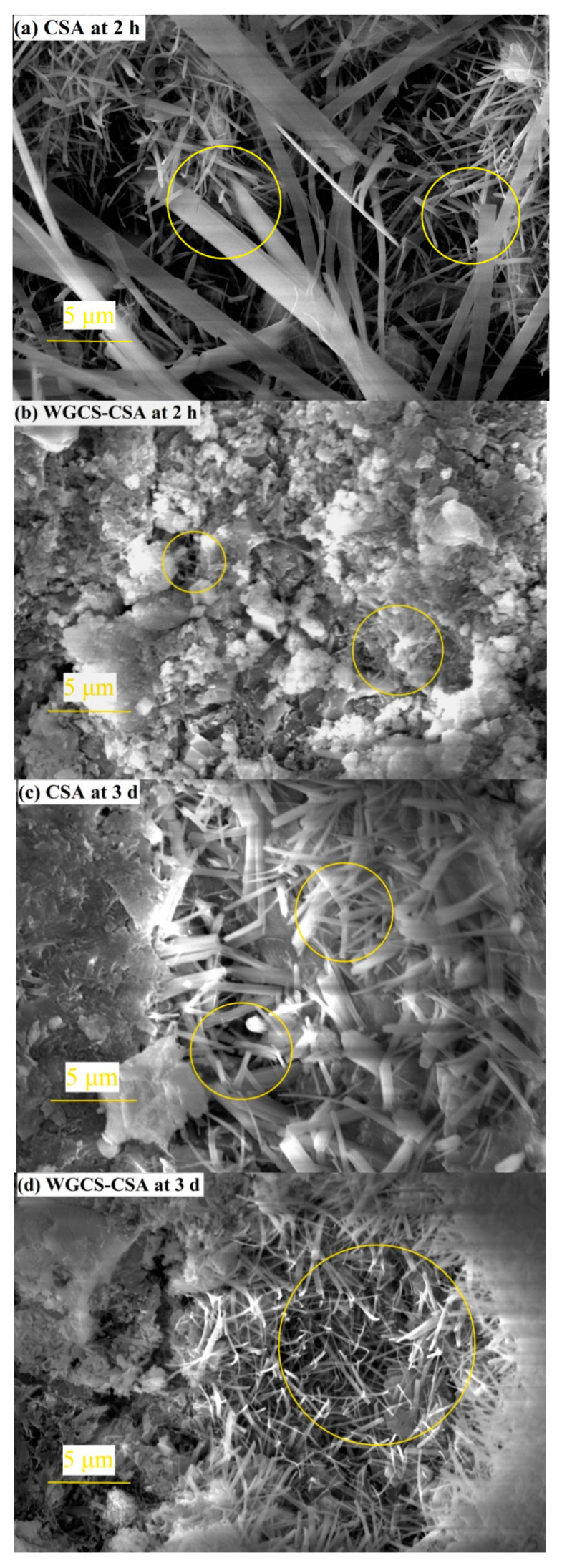
SEM images of CS–CSA paste. (**a**) CSA at 2 h; (**b**) WGCS-CSA at 2 h; (**c**) CSA at 3 d; (**d**) WGCS-CSA at 3 d.

**Figure 9 materials-13-04526-f009:**
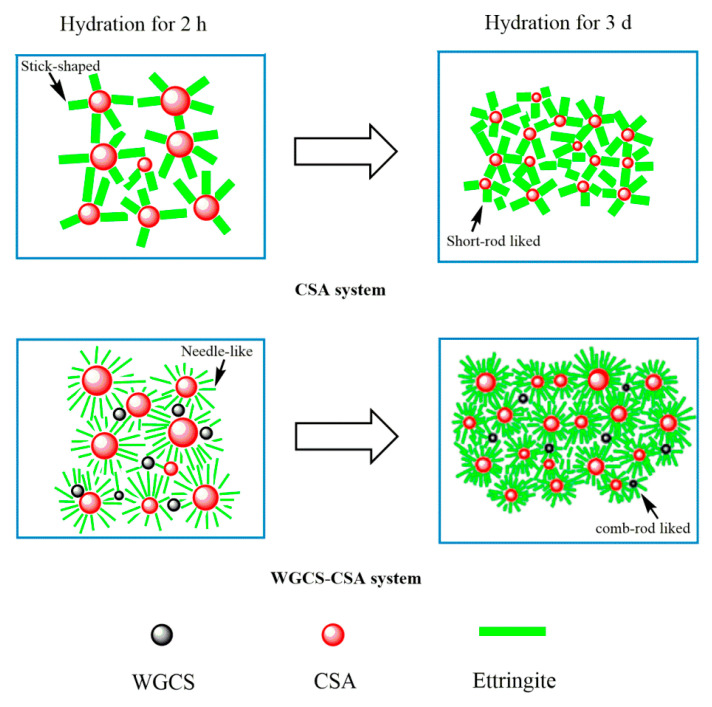
The mechanism of hydration in CSA and WGCS–CSA system.

**Table 1 materials-13-04526-t001:** Chemical compositions of calcium sulfoaluminate cement (CSA) and carbide slag (CS).

Compound	CaO	Al_2_O_3_	SiO_2_	SO_3_	MgO	Fe_2_O_3_	TiO_2_	K_2_O	Loss in Ignition (LOI)	Others
CSA	38.69	22.31	14.81	14.46	2.86	2.78	0.94	0.45	1.89	0.60
CS	64.09	1.62	2.90	0.27	0.10	0.11	0.03	–	30.23	0.65

**Table 2 materials-13-04526-t002:** Wet grinding parameters for the preparation of wet-ground carbide slag (WGCS).

RCS/g	Water/g	Dispersant/g	Zirconia Balls/g	Grinding Time/min	Grinding Speed/rpm
120	120	1.8	300	60	400

**Table 3 materials-13-04526-t003:** Mix designs of pastes.

NO.	CSA/g	WGCS Slurry/g	RCS/g	Water/g	PCE/g
1	2000	0	0	540	14
2	1920	160	0	460	14
3	1840	320	0	380	14
4	1760	480	0	300	14
5	1840	0	160	540	14

**Table 4 materials-13-04526-t004:** Setting time of CS–CSA pastes (maximum error was 2 min).

	Initial Setting Time (min)	Final Setting Time (min)
Blank	59	71
8% of RCS	13	16
12% of WGCS	16	19
8% of WGCS	14	16
4% of WGCS	12	14

**Table 5 materials-13-04526-t005:** Accumulated hydration heat of CS–CSA pastes from 10 to 120 min.

Age (min)	Blank (J/g)	RCS–CSA (J/g)	WGCS–CSA (J/g)
10	3.07	3.04	21.28
20	7.53	7.88	28.16
40	14.94	17.11	36.08
80	34.62	30.86	48.89
120	53.14	40.86	58.80

**Table 6 materials-13-04526-t006:** Content of ettringite in investigated specimens calculated by TG data (%).

Curing Age	Blank	WGCS–CSA
2 h	14.9	26.2
3 d	33.0	30.7

**Table 7 materials-13-04526-t007:** Cost of the materials.

Li_2_CO_3_ (CNY/t)	WGCS (CNY/t) [59]	CSA (CNY/t)
43,600	100	850

**Table 8 materials-13-04526-t008:** Cost of CSA and WGCS–CSA system.

System	Dosage of Li_2_CO_3_/WGCS (%)	Initial Setting Time Reduction Rate (%)	Final Setting Time Reduction Rate (%)	WGCS/Li_2_CO_3_ Cost (CNY/t)	Total Cost (CNY/t)
CSA + Li_2_CO_3_	0.03~0.09	25.0~30.0	20.5~28.0	4–39	863–888
WGCS–CSA	8	23.7	22.5	8	790

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
