# Peer review of "Utilization of Carbide Slag by Wet Grinding as an Accelerator in Calcium Sulfoaluminate Cement"

_materials, 2020, doi:10.3390/ma13204526_

Round 1

Reviewer 1 Report

Dear editor,

​This article is an analysis of the properties that affect the initial strength by testing wet-grinded carbide slag by replacement rate as an accelerator for calcium sulfoaluminate cement. It is a material that is cheaper than existing materials and can secure an initial strength equal to or higher than the existing material. Also, it is thought that it can be used for emergency construction.

However, I think there are some flaws and limitations to clarify the results, so it seems necessary to revise the following issues before publication.

1. Please explain the full name for the first abbreviation (e.g., GB, RMB, etc.) related to the overall article manuscript.

​2. Regarding the contents of the overall article, modify the chemical symbols (e.g., 4CaO·3Al2O3·SO3, Li2SO4, etc.) with subscripts and the units (e.g., mm3, etc.) with superscripts.

3. If the sum of chemical composition in Table 1 is less than 100%, insert “others” and fill in the remaining ratios.

4. It is necessary to correct a typo for “40℃” in Line 136. Please check overall manuscripts.

5. Why is the W/B not fixed at 0.27 in Table 3? Isn't it a test after fixing W/B constant?

6. Are there any test results for the flow of cement pastes?

7. In Figure 4, it is recommended to further explain why WGCS reduces compressive strength at the age of 3 and 7 days.

8. In Table 4, it is recommended to mention the relationship between powder fineness and setting time.

9. In Table 7, it is necessary to clarify the source of cost by materials.

10. In the conclusion, please reinforce the limitations of this study and future research directions.

Reviewer 2 Report

Thank you for your manuscript submission. I offer the following comments:

-In the introduction section, I suggest a brief introductory statement (1-2 sentences) that distinguishes CSA cements from normal Portland cement to help readers become more familiar with this type of material, with respect to its properties and composition.

-Please consider revising the grammar on statement on lines 52-55 as it is very long-winded (wordy)

-Please elaborate on the ‘nano effect’ mentioned in line 62

-The notation on the chemical formulas need to be formatted appropriately (e.g. instead of writing Ca(OH)2, write Ca(OH)2)

-What is CAC (line 71)? This hasn’t been spelled out before it is introduced as an acronym.

-The degree symbol is missing in several instances, please adjust accordingly (e.g. see line 109, line 136, line 257, line 259, etc.)

-How many measurements were taken to determine the particle size distribution? What is the mean and standard deviation?

-How was the particle size measured? Any specific software used?

-What is PCE? Please write the full name before using acronyms

-Notations on dimensions need to be written appropriately (e.g. line 131 on cubic millimeters)

-In section 2.3.1, there’s mention of statistical methods used – what statistical methods were used?

-Change ‘Loading speed’ to ‘Load rate’ in line 152.

-What is meant by “maximum number of images was 6144x4096” in line 172? Is this referring to the image’s resolution?

-Figure 4’s bar graphs should be visually easier to tell apart.

-What is the minimum desired strength for these materials? It would be helpful to include this as reference to the results obtained in this study.

-Should there be concerns that the strength of the WGCS was lower than the blank samples at ages 3d and 7d?

-Is there any reason as to why a higher % of WCGS increases the set time slightly?

-What is meant by the ‘covering effect’ in line 226? Please elaborate

-What is the permeability of these type of cements? This paper seems to suggest compressive strength is the only material property that matters in construction – there should be a statement clarifying that while this material shows promise in rapid strength, it is not the only parameter that needs to be looked at for durable construction (and that maybe this could be future work?).

-In the conclusion statement, remove the parenthesis in line 341.

Reviewer 3 Report

Title: Utilization of carbide slag by wet grinding as an accelerator in calcium sulfoaluminate cement

General notes for work:

The paper lacks information on how many samples were used in the strength test and how many attempts were made to measure the setting time. This is important information because of the basic objective of the experiment. It is important to demonstrate the reproducibility of the results, and this is possible with a large number of measurements.

I recommend publishing the article after supplementing the amount of samples in the test of strength and setting time

Suggestion beyond review. Have the authors considered using alcohol instead of water in the carbide slag grinding process?

Round 2

Reviewer 1 Report

The answer to the question was appropriate. It is judged that it is ready for publication.

This manuscript is a resubmission of an earlier submission. The following is a list of the peer review reports and author responses from that submission.

Round 1

Reviewer 1 Report

This study dealt with interesting topic that used WGCS as an accelerator in CSA system. However, some major and minor comments should be considered before publication.

1. In abstract, the phrase “2h compressive strength by above 6 times” is recommended to revised to specify the exact strength figure not the multiple.

2. In Introduction section, please cite and include the following papers as the references in the first paragraph. They deal with CSA cement systems modified by redispersible polymer and set retarders.
1) Gwon, S., Jang, S. Y., & Shin, M. (2018). Combined effects of set retarders and polymer powder on the properties of calcium sulfoaluminate blended cement systems. Materials, 11(5), 825.
2) Gwon, S., Jang, S. Y., & Shin, M. (2018). Microstructure evolution and strength development of ultra rapid hardening cement modified with redispersible polymer powder. Construction and Building Materials, 192, 715-730.

3. In Introduction section, conventional accelerators like lithium salts are introduced. It would be better to compare how cheap or expensive the CS is compared with commercial accelerators.

4. In Section 2.2.1, the process of WGCS production is introduced. It seems that this process is quite expensive to produce final product. Are the authors sure that WGCS is more economical than commercial accelerators? If so, please give some persuasive rationales.

5. In Table 2, the wet grinded parameters for WGCS is provided. Only one type of WGCS is considered in this study. Is this an optimal design? Please explain why this design is selected in this study. Also, there is a typo, WGWC in Table 2.

6. Please upgrade the quality of all the figures in the manuscript. Some figures (particularly SEM) exhibit a bad resolution.

7. The representation of PDF numbers of portlandite and calcite should be revised to include “ICDD”. Please cite the data base (ICDD) as the reference.

8. Please briefly describe how the SEM images of RCS and WGCS were obtained using SEM in Section 2.2.1.

9. Please confirm that the maximum XRD peaks of RCS and WGCS are matched. Because the XRD intensity is not provided, at least, the maximum peak (around 29 2 theta) of calcite should be same in both XRD patterns.

10. Please revise Table 3 to show the mix proportions in kg/m3.

11. Why were the compressive strengths measured up to 7 d? Why not 28 d strength?

12. In Section 2.3.3, the scanning speed of XRD was set at 4 degree/min. This seems relatively speedy. Why is this speed chosen? 1 degree/min of scanning is commonly used for XRD.

13. In Section 2.3.5, please describe more about test conditions including electron beam. You may the two suggested papers in no.2 comment.

14. Please indicate “WGCS” in Fig. 1(a).

15. Again, why were the compression tests performed only up to 7 days? Excluding the compressive strength at 2 h, the mixtures including RCS and WGCS always achieved smaller strengths than Blank. Do you expect that the mixtures including RCS and WGCS will develop higher strengths than Blank at 28 d?

16. In Section 2.3.1, please cite the reference of setting time test standard and describe more about test method.

17. For making SEM sample, how is the sample ground? By pestle?

18. In Fig. 1(a), why did the use of 12% WGCS induce a strength decrease at 7 d compared to 3 d?

19. Regarding Fig. 2, rather than showing the setting time data in figure, please use table to show both initial and final setting times clearly.

20. In Section 3.3, only two mixtures are tested and analyzed. Why are the other mixtures not tested? For example, isothermal calorimetry on 8% RCS will be helpful for comparison with Blank and 8% WGCS cases. Current data just show only two cases. Thus, the Reviewer ask the authors to conduct test at least 8% RCS case.

21. In Fig. 6(a), both Blank and 8% WGCS have almost the same first peak with similar heat rate. Even though whether each peak originate from what kind of chemical reaction is described in Section 3.3, it would be more helpful to indicate such information in Fig. 6(a). In addition, please cite and add relevant papers that confirm the information in the text. Current description in Section 3.3 does not include any reference.

22. In line 260 to 261 at page 10, “The main reason was that the presence of CS greatly promoter the pH value of the system” appeared to mean that pH value was measured in this study. If it’s not, please remove or revise this sentence.

23. For removing water from SEM sample, ethanol was used. Isopropanol is usually used for hydration stoppage. Is there a relevant paper that the authors followed this hydration stoppage method? If so, please add it as a reference in Section 2.2.2. Hydration stoppage method is very critical to secure the microstructure of cement paste sample at a certain age. Please explain the hydration stoppage method in more detail including volumetric ratio of solid sample-to-ethanol. The following paper may contain a round robin test of hydration stoppage.
1) Snellings, R., Chwast, J., Cizer, Ö., De Belie, N., Dhandapani, Y., Durdzinski, P., ... & Santhanam, M. (2018). Report of TC 238-SCM: hydration stoppage methods for phase assemblage studies of blended cements—results of a round robin test. Materials and Structures, 51(4), 111.